# Incidence and Characteristics of Kidney Stones in Patients on Ketogenic Diet: A Systematic Review and Meta-Analysis

**DOI:** 10.3390/diseases9020039

**Published:** 2021-05-25

**Authors:** Prakrati Acharya, Chirag Acharya, Charat Thongprayoon, Panupong Hansrivijit, Swetha R. Kanduri, Karthik Kovvuru, Juan Medaura, Pradeep Vaitla, Desiree F. Garcia Anton, Poemlarp Mekraksakit, Pattharawin Pattharanitima, Tarun Bathini, Wisit Cheungpasitporn

**Affiliations:** 1Division of Nephrology, Texas Tech Health Sciences Center El Paso, El Paso, TX 79905, USA; 2Lea County Correctional Facility, Hobbs, NM 88240, USA; drchiragacharya1985@gmail.com; 3Department of Medicine, Mayo Clinic, Division of Nephrology and Hypertension, Rochester, MN 55905, USA; 4Department of Internal Medicine, UPMC Pinnacle, Harrisburg, PA 17105, USA; hansrivijitp@upmc.edu; 5Department of Medicine, Ochsner Medical Center, New Orleans, LA 70121, USA; svetarani@gmail.com (S.R.K.); karthikreddy.999@gmail.com (K.K.); 6Division of Nephrology, Department of Internal Medicine, University of Mississippi Medical Center, Jackson, MS 39216, USA; jmedaura@umc.edu (J.M.); pvaitla@umc.edu (P.V.); dgarciaanton@umc.edu (D.F.G.A.); 7Department of Internal Medicine, Texas Tech University Health Science Center, Lubbock, TX 79430, USA; poemlarp@gmail.com; 8Department of Internal Medicine, Faculty of Medicine, Thammasat University, Pathum Thani 10120, Thailand; 9Department of Internal Medicine, University of Arizona, Tucson, AZ 85721, USA; tarunjacobb@gmail.com

**Keywords:** ketogenic diet, kidney stones, nephrolithiasis, epidemiology, meta-analysis, systematic review

## Abstract

Very-low-carbohydrate diets or ketogenic diets are frequently used for weight loss in adults and as a therapy for epilepsy in children. The incidence and characteristics of kidney stones in patients on ketogenic diets are not well studied. **Methods:** A systematic literature search was performed, using MEDLINE, EMBASE, and Cochrane Database of Systematic Reviews from the databases’ inception through April 2020. Observational studies or clinical trials that provide data on the incidence and/or types of kidney stones in patients on ketogenic diets were included. We applied a random-effects model to estimate the incidence of kidney stones. **Results:** A total of 36 studies with 2795 patients on ketogenic diets were enrolled. The estimated pooled incidence of kidney stones was 5.9% (95% CI, 4.6–7.6%, I2 = 47%) in patients on ketogenic diets at a mean follow-up time of 3.7 +/− 2.9 years. Subgroup analyses demonstrated the estimated pooled incidence of kidney stones of 5.8% (95% CI, 4.4–7.5%, I2 = 49%) in children and 7.9% (95% CI, 2.8–20.1%, I2 = 29%) in adults, respectively. Within reported studies, 48.7% (95% CI, 33.2–64.6%) of kidney stones were uric stones, 36.5% (95% CI, 10.6–73.6%) were calcium-based (CaOx/CaP) stones, and 27.8% (95% CI, 12.1–51.9%) were mixed uric acid and calcium-based stones, respectively. **Conclusions:** The estimated incidence of kidney stones in patients on ketogenic diets is 5.9%. Its incidence is approximately 5.8% in children and 7.9% in adults. Uric acid stones are the most prevalent kidney stones in patients on ketogenic diets followed by calcium-based stones. These findings may impact the prevention and clinical management of kidney stones in patients on ketogenic diets.

## 1. Introduction

The ketogenic diet, initially introduced in the early nineteenth century, refers to a diet pattern that is low in carbohydrates and high in fat with a moderate proportion of protein (1.2–1.5 g/kg) [1,2]. The ketogenic diet increases the oxidation of fatty acids and ketone bodies production—creating a state of ketosis and mild acidosis [3,4]. A ketogenic diet leads to glycolysis inhibition, inhibits glutamatergic synaptic transmission, and assists in weight loss [5,6], making it popular not only for patients with obesity or metabolic syndrome, but even for athletes, both professional and amateur [7]. Ketone bodies (acetate, aceto-acetate, and beta-hydroxybutyrate) have been shown to prevent recurrent seizures [8,9,10], hence are prescribed for children with intractable seizures. However, the mechanism of the anti-seizure effects of ketone bodies is not well understood. The ketogenic diet may also have a protective effect against cognitive impairment [11] and malignancy [12]. Indications for the ketogenic diet have been extended to include glucose-1 transporter deficiency syndrome and pyruvate dehydrogenase deficiency disorders [13,14].

Multiple formulations of the ketogenic diet are currently available, including the classic keto diet, low glycemic index diet (LGID), medium-chain triglyceride diet (MCT), and modified Atkins diet. These diets differ in the proportions of lipid, carbohydrate, and protein contents [10,13,14,15]. Despite potential advantages, the ketogenic diet has multiple adverse effects. During the first four-week period, nausea, vomiting, and diarrhea are particularly common with the medium-chain triglyceride diet [16,17,18], posing a risk for acute kidney injury, hyponatremia, hypomagnesemia, hypercalciuria, hyperuricemia, and metabolic acidosis [19,20,21]. Long-term adverse effects of the ketogenic diet, including osteopenia, risk of bone fractures, alterations in vitamin D levels, are well reported [22,23,24]. Increased risk for kidney stones is well described in patients using the ketogenic diet for over a 2 year period [17,20,25,26], with complications such as obstructive uropathy, acute kidney injury, and chronic kidney disease [27,28,29].

The incidence of kidney stones among patients on the ketogenic diet ranges from 3% to 10% [20,30,31], compared to one in several thousand in the general population [17,24]. We performed a meta-analysis on the incidence and characteristics of kidney stones in patients on the ketogenic diet to better understand the kidney stones’ burden and pathophysiology in this population.

## 2. Materials and Methods

### 2.1. Search Strategies

A comprehensive search of several databases from each database’s inception to 3 May 2019 was conducted. The databases included OVID MEDLINE (1946 to April 2020), EMBASE (1988 to April 2020), and the Cochrane Database of Systematic Reviews (database inception to April 2020). The systematic literature review was conducted independently by two investigators (P.A. and C.A.), using the search strategy that consolidated the terms of (‘ketogenic diet’ OR ‘keto diet’ OR ‘atkins diet’ OR ‘low carb diet’ OR ‘low carbohydrate diet’) AND (nephrolithiasis OR ‘kidney stone’ OR ‘kidney stones’). The actual strategy listing all search terms used is available in the online Appendix A. There were no restrictions on language, sample size, or study duration. This study was conducted by the Preferred Reporting Items for Systematic Reviews and Meta-Analysis (PRISMA) statement (online Appendix A) [32].

### 2.2. Study Selection

Eligible studies must be clinical trials, observational studies (cohort, case-control, or cross-sectional studies) that reported incidence and characteristics of kidney stones in patients on ketogenic diets. Retrieved articles were individually reviewed for eligibility by the two investigators (P.A. and C.A.). Discrepancies were addressed and resolved by third investigator (W.C.). Inclusion was not limited by language, age, sample size, or study duration.

### 2.3. Data Extraction

The following data were extracted: first author name, year of publication, number of patients, duration of follow-up, description of ketogenic diet, mean age, sex, incidence of kidney stones, type of kidney stones, and time to diagnosis of kidney stones after ketogenic diet consumption. The primary outcome included the incidence of kidney stones.

### 2.4. Data Synthesis and Statistical Analysis

We calculated the pooled estimated incidence of kidney stones among patients on the ketogenic diet. The pre-specified subgroup analysis based on age groups (pediatrics and adults) was performed. A random-effects model was used due to the expected clinical heterogeneity in the included populations [33]. All pooled estimates were shown with 95% confidence intervals (CIs). Heterogeneity among effect sizes estimated by individual studies was described with the I^2^ statistic and the chi-square test. A value of I^2^ of 0% to 25% represents insignificant heterogeneity, 26% to 50% low heterogeneity, 51% to 75% moderate heterogeneity and 76% to 100%, high heterogeneity [34].

Publication bias was evaluated using the Egger test [35]. A *p*-value of less than 0.05 indicates the presence of publication bias. The meta-analysis was performed by the Comprehensive Meta-Analysis 3.3 software (Biostat Inc, Englewood, NJ, USA). The data for this meta-analysis are publicly available through the Open Science Framework on 25 September 2020 (URL: https://osf.io/2gtk3/ (accessed on 25 September 2020)).

## 3. Results

A total of 221 potentially relevant articles were identified and screened. Fifty-one articles were assessed in detail of which 36 studies with 2795 patients on ketogenic diets were enrolled in our meta-analysis (Figure 1 and Table 1). The definitions and reported adverse effects observed in ketogenic diets in different studies are shown in Table 2.

### 3.1. Incidence of Kidney Stones among Patients on Ketogenic Diets

The estimated pooled incidence of kidney stones was 5.9% (95% CI, 4.6–7.6%, I^2^ = 47%, Figure 2) in patients on ketogenic diets at a mean follow-up time of 3.7 +/− 2.9 years. Subgroup analyses demonstrated the estimated pooled incidence of kidney stones of 5.8% (95% CI, 4.4–7.5%, I^2^ = 49%) in children and 7.9% (95% CI, 2.8–20.1%, I^2^ = 29%) in adults (Figure 3), respectively.

### 3.2. Type of Kidney Stones among Patients on Ketogenic Diet

Within reported studies, 48.7% (95% CI, 33.2–64.6%) of kidney stones were uric stones, 36.5% (95% CI, 10.6–73.6%) were calcium-based (CaOx/CaP) stones, and 27.8% (95% CI, 12.1–51.9%) were mixed uric acid and calcium-based stones, respectively.

### 3.3. Evaluation for Publication Bias

Using Egger’s regression asymmetry tests, there was no significant publication bias found in this meta-analysis. The Egger’s regression test demonstrated no significant publication bias in all analyses (*p* > 0.05).

## 4. Discussion

Our analysis reports a pooled incidence of kidney stones at 5.6% in patients treated with a ketogenic diet after four years. The incidence of nephrolithiasis in the general population is reported at 0.3% per year in men and 0.25% per year in women [72]. In our study, the incidence of kidney stones is identical in children and adults. This finding contradicts the hypothesis that children are more susceptible to kidney stone formation due to extended treatment duration with the ketogenic diet, small renal tubular lumen, and relatively less renal reserve. However, studies included in this analysis did not report the recurrence of kidney stones; it is possible children may be predisposed to recurrent kidney stones due to prolonged exposure to the ketogenic diet. More studies are required to understand the risk of recurrent kidney stones with the ketogenic diet.

Uric acid stones are the most common stones in patients receiving the ketogenic diet, followed by calcium-based stones and uric acid–calcium mixed stones. In contrast, calcium oxalate stones are the most common stones in the general population [72]. The exact mechanism of nephrolithiasis following the ketogenic diet is unclear. However, it is likely related to hypocitraturia and acidosis, common in people consuming a high-protein and low-alkali diet [20]. Acidosis contributes to significant reabsorption of citrate in the proximal tubule, further contributing to hypocitraturia [25,73,74,75,76,77,78]. A more generous amount of free calcium is available for stone formation in a low-citrate environment [73,79]. Chronic acidosis also leads to demineralization of the bone and increased calcium excretion [17,20]. Hypercalciuria, immobilization, anti-epileptic drugs, and fat malabsorption further precipitate urinary calcium. Moreover, the low urine pH seen in patients with a low-alkali diet contributes to uric acid crystals [73]. Obesity, insulin resistance, and an animal-protein diet are associated with low urine pH [80]. The uric acid stone may act as a nidus for calcium-based nephrolithiasis formation [73]. Furthermore, fluid intake restriction is traditionally applied to children receiving the ketogenic diet, making them susceptible to stone formation [73].

Potential benefits of urine alkalization with oral potassium citrate in children with a urine calcium to creatinine ratio of >0.20 mg/mg to prevent kidney stone formation is well reported [20,25,75,78]. Genetic polymorphisms in transporters, such as renal sodium citrate cotransporter, is a known risk factor in recurrent stone formers [81,82]. McNally et al. reported that the empiric use of oral citrate in children treated with a classic ketogenic diet led to a reduction in the incidence of kidney stones from 6.75% to 0.9% without an increase in adverse effects [67]. The international ketogenic diet study group agreed that oral citrates appear to prevent kidney stones; however, there was mixed opinion on its empiric use (class III) [9]. The consensus statement is unchanged since 2009 due to the lack of new evidence. Hence, we need a well-designed study to analyze the empiric use of urine alkalization therapy. A ketogenic diet is generally prescribed for weight loss in adults, who require a shorter duration of therapy; the empiric use of oral citrates may not be necessary. However, this remains to be elucidated by future studies.

Purine-rich foods (red meat, fish, poultry, beer, and legumes) increase the uric acid load [80]. The digestion of animal protein produces a transient acidic environment, which results in a lower urine pH, promoting the precipitation of uric stones [80]. Since uric stones are the most common stones in patients receiving a ketogenic diet, switching from animal proteins to plant-based proteins results in lower uric acid excretion, but, sequentially, lower uric acid stone formation is unclear. Siener et al. reported that patients consuming a balanced diet of vegetables and animal proteins had higher urine pH and urine uric acid concentration than those on a typical western diet [83]. It is also recommended that patients with symptomatic hypercalcemia, hyperuricosuric calcium urolithiasis, and urate nephropathy should be prescribed a urate-lowering agent [80]. However, empiric use of xanthine oxidase inhibitors in patients on a ketogenic diet requires further investigation.

Other measures to mitigate the risk of renal stones include liberalizing fluid intake and avoiding the initial fasting phase at the start of ketogenic diets [84,85]. Modification of the diet regimen to allow small, frequent meals might help decrease gastrointestinal side effects and avoid volume depletion [86]. Screening for underlying metabolic disorders should be considered before initiation of a ketogenic diet to help avoid substantial acidosis [87]. Considering the long-term risk of bone fractures and osteopenia, the 2018 international ketogenic diet group recommended periodic DEXA scan screening for evaluation of bone mineral density [9]. Epidemiological studies have shown a temporal relationship between idiopathic osteoporosis and kidney stones. In addition, changing dietary patterns, including the ketogenic diet, could possibly be an important environmental trigger in the association, as well [88]. Bone health should be monitored closely in patients on the ketogenic diet and more clinical trials are needed to further define the negative impacts on bone health [89]. Although prophylactic calcium and vitamin D is recommended in all people on the ketogenic diet for bone health [23,24], athletes with dermal calcium loss during exercise/sweating and obese subjects restricting dairy are at further risk of worsening bone health if not on adequate calcium supplements [7,89]. However, given the reported risk of nephrolithiasis from hypercalciuria, supplementation remains a challenge [20,25]. Periodic urine chemistry analyses to measure the calcium to creatinine ratio, calcium, citrate, and oxalate can help identify patients at risk for kidney stone formation, and timely referral to a nephrologist should be considered. Patients with a family history of nephrolithiasis should be screened before starting a ketogenic diet due to their higher risk for stone formation.

In the era of precision medicine, further studies are needed to understand the use of genetic variants to further personalize management, even in the dietary therapy field [90]. Our study has the following limitations. First, the observational studies included in the analysis are susceptible to shortcomings inherent to the design. In addition, sources of heterogeneity could be due to differences in patient population and definitions of ketogenic diet as described in Table 2. Second, important risk factors, such as family history of nephrolithiasis, physical activity, exposure to sunlight, presence of ketone bodies in the blood or urine and environmental exposure, were not mentioned in the included studies. Third, recurrent kidney stones are not differentiated from the first episodes; recurrent kidney stones might be more common in children. Fourth, the glomerular filtration rate decline was not the primary outcome in most of the included studies. Fifth, the pooled sample size for adults is smaller than children. Lastly, data on the estimated incidence of kidney stones in elderly and CKD patients on ketogenic diets were limited.

## 5. Conclusions

In conclusion, the estimated incidence of kidney stones in patients on ketogenic diets is 5.6%, which is comparable among adults and children. Uric acid stones are the most prevalent kidney stones in patients treated with ketogenic diets, followed by calcium-based stones. These findings may impact the prevention and management of kidney stones in patients treated with ketogenic diets.

## Figures and Tables

**Figure 1 diseases-09-00039-f001:**
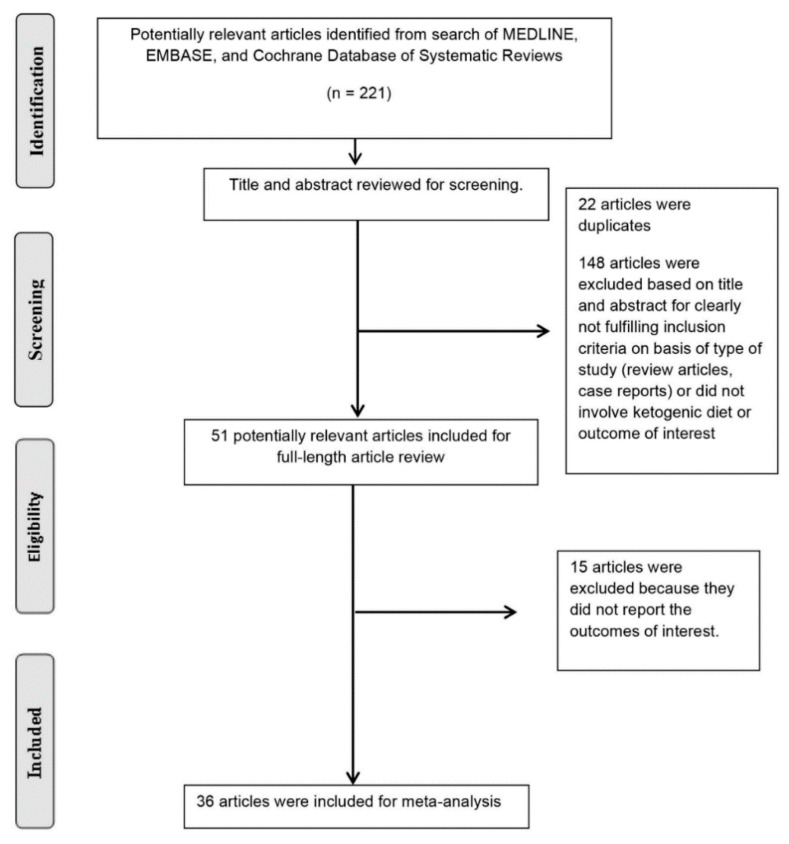
The literature retrieval, review, and selection process.

**Figure 2 diseases-09-00039-f002:**
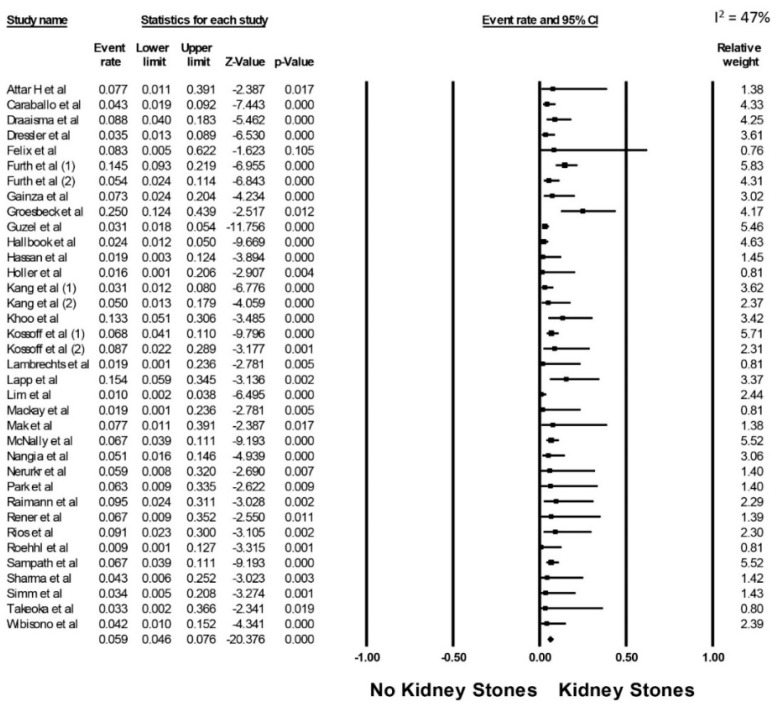
Pooled estimated incidence of kidney stones.

**Figure 3 diseases-09-00039-f003:**
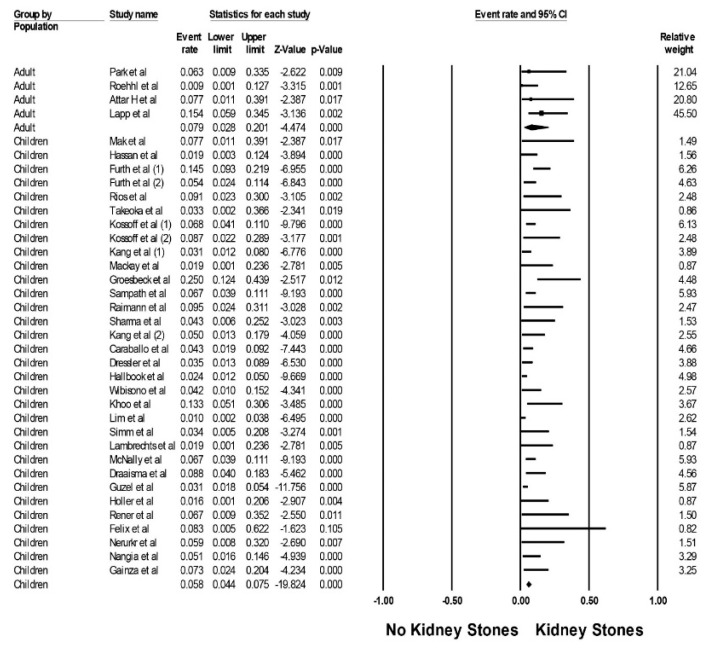
Pooled estimated incidence of kidney stones by patient population.

**Table 1 diseases-09-00039-t001:** Characteristics of studies included in this systematic review.

Reference	Description of Ketogenic Diet	Total Number of Patients on Ketogenic Diet	Mean Age of Included Patients	Mean Duration of KGD	No. of Patients with Kidney Stones	% of Patients with Kidney Stones	24 h Urine Study
Holler A. et al. [36]	Classical KD (16), Modified Atkins diet (MAD) (14)	31	Median 5.5, mean 5.5 (0.1–15.7 Y)	N/A	0	0	Hypercalciuria in 12/22
Rener et al. [37]	2.5:1 to 45/3/20201	15	<3 Y	Average 13 M (4–16 M)	1	6.60%	N/A
Attar H. et al. [38]	Modified Atkins diet	13	23–72 Y	Range 1–21 M	1	7.69%	N/A
Felix et al. [39]	Modified Atkins diet	5	6–12 Y	4 M	0	0	1 had hypercalciuria
Lapp et al. [40]	KGD in 7 and Modified Atkins diet in 19	26	>18 Y	N/A	4	15%	N/A
Nerurkar et al. [41]	KGD	17	<3 Y	N/A	1	5.88%	N/A
Nangia et al. [42]	KGD	59	4.5 Y (range 0.2–22 Y)	Mean 2 Y (range 0–5.5 Y)	3	5%	N/A
Gainza et al. [43]	KGD	41	4.7 Y (range 1–13 Y)	Mean 5.79 Y (3–10.6)	3	7.10%	Hypercalciuria in 11 (26.2%)
Mak et al. [44]	KGD (MCT oil diet)	13	7.3 Y	N/A	1	7.70%	Increase Ca/Cr ratio in 1 patient
Furth et al. [45]	N/A	112	5 Y	N/A	6	5.30%	Elevated Ca/Cr ratio
Rios et al. [46]	4:1 (1.5:1–4.5:1) KGD	22	Range 1–19 Y (min age 18 M)	25 M (1–54 M)	2	9.09%	N/A
Sharma et al. [47]	3:1 KGD in <18 M, 4:1 KGD in >18 M	23	(range 6 M–5 Y)	N/A	1	3.70%	Elevated Ca/Cr ratio
Kang et al. [48]	3:1 KGD	40	Median ± IQR- 15 ± 13.0 (range 6–60 M)	N/A	2	5.00%	N/A
Wibisono et al. [49]	Classic KGD, MCT diet, Modified Atkins diet	48	3.8 Y (range 2.3–7 Y)	Range 1–14 Y	2	4.00%	N/A
Simm et al. [50]	4:1 to 2:1 KGD	29	6.4 Y (range 3.3–17.8 Y)	Mean 2.1 Y (range 0.5–6.5 Y)	1	N/A	N/A
Guzel et al. [51]	Olive oil-based KGD	389	Median 4.0 (2–7)Y	12 M	12	3%	N/A
Hassan et al. [52]	Classic 4:1 KGD (49 of 52 pts), rest with modified diet supplemented by MCT oil	52	5 Y, 6 M ± 3 Y, 4 M	N/A	1	1.90%	Increased calcium oxide in urine
Takeoka et al. [53]	3:1 to 4.1 KGD + Topiramate	14	Mean age 4.7 Y	N/A	0	0%	N/A
Kossoff et al. [54]	KGD (older children on 4:1 diet and younger on 3:1) without carbonic anhydrase inhibitors	221	5.1 Y (SD 4.5, range 16.5 Y)	N/A	15	6.70%	N/A
Kossoff et al. [55]	KGD (older children on 4:1 diet and younger on 3:1) with topiramate or zonisamide	80	4.8 Y (SD 2.4, range 6.5 Y)	N/A	5	6.30%	N/A
Kossoff et al. [56]	4:1 KGD in 9, 3.5:1 in 1 and 3:1 in 13	23	1.1 Y (range 0.5–24 M)	N/A	2	8.60%	N/A
Kang et al. [57]	4:1 KGD	129	64.9 (±59.3) M	12.0 (±10.1) M	4	3.10%	N/A
Mackay et al. [58]	N/A	26	Median age 6.1 Y (range 2.3–13.2)	N/A	0	0%	Increased urine calcium in 8%
Groesbeck et al. [59]	4:1 KGD in 19, 3:1 KGD in 9	28	3 Y 9 M (range 6 M–13 Y 6 M)	7 Y 9 M	7	25%	Increase Ca Cr ratio in 14 pts
Sampath et al. [60]	3:1 (56%) or 4:1 KGD	195	Median 3 Y (0.5–15 Y)	Median 12 M (range 1–72 M)	13	7%	N/A
Raimann et al. [61]	4:1 in 16, 3.5:1 in 3 and 3:1 in 2 + Calcium and MV supplement	21	6.2 Y (range 6 M–17 Y)	15 pt completed 1 Y of KGD 2.6 Y (1–6.3 Y)	2	10%	Hypercalciuria in both stone formers
Caraballo et al. [62]	N/A	140	5 Y (range 1–18 Y	3.5 Y (range 1–20 Y)	6	4.28%	N/A
Dressler et al. [63]	4:1 in 36, 3:1 in 53, 3.5:1 in 6, 2.5:1 in 17, 2:1 in 3	115	2.86 ± 3.1 (min 0.0–max 16.8)	N/A	4	3.40%	N/A
Hallbook et al. [64]	N/A	290	5.3 (0.6–18.6)	2 Y	7	N/A	N/A
Khoo et al. [65]	4:1 in 12, 3:1 in 11, 2:1 in 3, MAD in 4	30	6.8 Y (8 M to 17 Y)	8 M (range 7 days to 6 Y)	4	13%	N/A
Lim et al. [66]	N/A	204	4.8 Y (range 0.3–33.9 Y)	Median 17 M (95% CI 9–24 M)	2	0.98%	N/A
McNally et al. [67]	3:1 or 4;1 KGD	195 (KGD + hypercalciuria so polycitra K given)	4.3 Y in reactive group	15.6 (13.1)	13	6.70%	N/A
Park et al. [68]	N/A	16	Age range (0.1–40 Y)	N/A	1	6.25%	N/A
Draaisma et al. [69]	Classic KGD (67.6%), MCT diet (2.9%), MAD (19.1%) or LGIT (7.4%), other (1.5%)	68	5.7 ± 4.3 Y	25.6 ± 24.8 M	6	8.80%	N/A
Roehhl et al. [70]	Modified KGD	55	Mean 38 Y (range 17–70 Y)	N/A	0	0%	N/A
Lambrechts et al. [71]	MCT diet and Classic KGD	26	7 Y	0	0	N/A	N/A

Abbreviations: KGD—ketogenic diet, MCT—medium chain triglyceride, F/H—family history, Y—years, M—months, pts—patients, Ca—calcium, Cr—creatinine, MV—multivitamin, K citrate—potassium citrate, RR—relative risk, NR—not recorded, gp—group, LGIT—low glycemic index treatment, MAD—modified Atkin’s diet.

**Table 2 diseases-09-00039-t002:** Definitions and reported adverse effects observed in ketogenic diets in different studies.

Author	Different Types of Ketogenic Diet	Side Effects/Complication of Ketogenic Diet besides Renal Stones
Holler A. et al. [36]	Classical KGD and MAD	Constipation, increased bromine level (3.2%)
Rener et al. [37]	KGD 2.5:1 to 4:1	Vomiting
Attar H. et al. [38]	MAD	NR
Felix et al. [39]	MAD	Weight loss, hyperlipidemia
Lapp et al. [40]	KGD, MAD	Gallstones (3.8%), hyperlipidemia (3.8%)
Nerurkar et al. [41]	KGD not specified	Constipation (57%)
Nangia et al. [42]	KGD 3:1 to 4:1	Constipation (39%), acidosis (21%), nausea/emesis (14%), increased seizures (7%).
Gainza et al. [43]	KGD not specified	Osteopenia (38.1%), severe metabolic acidosis (9.5%), recurrent pneumonia (21.4%), neutropenia (0.5%), fatty liver (0.1%), easy bruising (4.8%)
Mak et al. [44]	KGD—Protein + carb (<19%) of caloric requirements MCT 60–70% of caloric requirements	Weight loss (46%), diarrhea (38%), bad temper (7.6%), abdominal cramps (15%), nausea (15%), bad body smell (7.6%)
Furth et al. [45]		NR
Rios et al. [46]	KGD 4:1 (1.5:1 to 4.5:1)	Nausea and vomiting (26.3%), hypercholesterolemia (64.7%), anorexia (31.8%), constipation (40.9%), symptomatic acidosis (9.09%), carnitine deficiency (9.09%)
Sharma et al. [47]	Classical KGD 3:1 or 4:1	Vomiting (75%), asymptomatic hypocalcemia, Constipation (75%), weight loss, hypoalbuminemia
Kang et al. [48]	Classical KGD 4:1	Dehydration, GI discomfort, hyperlipidemia, hyperuricemia, symptomatic hypoglycemia, lipoid aspiration pneumonia, hypoproteinemia, hypomagnesemia, repeated hyponatremia
Wibisono et al. [49]	Classical KGD, MCT, MAD	Constipation, hypertriglyceridemia, hypercholesterolemia, diarrhea, lethargy, iron deficiency, GERD, vomiting, hypoglycemia
Simm et al. [50]	KGD 2:1 to 4:1	Osteopenia, fracture
Guzel et al. [51]	KGD 2.5:1 and 4:1	Hyperlipidemia (50.8%), selenium deficiency (26.9%), constipation (26.2%), sleep disturbances (20%), hyperuricemia (3%), hepatic effects (2.6%), hypoproteinemia (2.6%), hypoglycemia(1.5%)
Hassan et al. [52]	Classic 4:1 KGD or MCT diet	Constipation (85%), gall bladder stone (1.9%), hyponatremia (1.9%)
Takeoka et al. [53]	KGD not specified	Nausea/vomiting (7%), irritability (7%), lethargy (21%), sedation (14%)
Kossoff et al. [54]	KGD 3:1 to 4:1	NR
Kossoff et al. [55]	KGD 3:1 to 4:1	Sedation (27%), rash, irritability
Kossoff et al. [56]	KGD 3:1 to 4:1	Severe GERD (13%), hip dislocation (0.4%)
Kang et al. [57]		Dehydration, GI discomfort, hyperlipidemia, hyperuricemia, hypoglycemia
Mackay et al. [58]	Classical KGD 3:1 to 4.2:1	Asymptomatic hypoglycemia (24%), poor linear growth (20%), hyperlipidemia (16%), vomiting (12%), hypocarnitinemia (8%), hypercalciuria (8%), constipation (8%), osteopenia (4%), pancreatitis (4%), Diarrhea (4%)
Groesbeck et al. [59]	60.7% on classical KDT 7% MAD, 32% other KGD	Fractures (21.4%), hyperlipidemia (7%), constipation (53%)
Sampath et al. [60]	KGD 3:1 (56%) or 4:1	NR
Raimann et al. [61]	KGD 4:1 (3 pts 3.5:1 2 pts 3:1)	Hypercholesterolemia 64% (at 12 months) 15% (at 18 months), growth retardation
Caraballo et al. [62]	KGD	GI side effects (30.5%), hyperlipidemia (9.7%), weight gain (2.3%), hypocarnitinemia (3.7%), hypercalciuria (6.9%), hypoglycemia (5.5%), dehydration (6.4%)
Dressler et al. [63]	KGD 3:1, 4:1 or 2.5:1	Carnitine deficiency (13%), growth deficit (5.2%), weight gain (1.7%), hypertriglyceridemia (29.5%), hypercholesterolemia (10.4%)
Hallbook et al. [64]	KGD 3:1 or 4:1 ratio	Hyperlipidemia (6%), bone fractures (0.9%)
Khoo et al. [65]	Classical KGD (81.2%), MAD (18.75%)	Constipation (43%), hunger (23%), excessive weight gain or loss (20%), vomiting (10%), hyperuricemia (30%), hypocalcemia (20%)
Lim et al. [66]	NR	GI side effects (nausea, vomiting, and constipation), Inadequate weight gain or significant weight loss, ketoacidosis, hepatotoxicity, renal dysfunction, sinus tachycardia, osteoporosis
McNally et al. [67]	KGD unspecified	NR
Park et al. [68]	KGD 4:1 (87.5%), KGD 3:1 (12.5%)	Regurgitation, constipation, aspiration, hypertriglyceridemia, hypoproteinemia, nausea, vomiting
Draaisma et al. [69]	Classic KGD (67.6%), MCT diet (2.9%) MAD (19.1%), LGID (7.4%), others (1.5%)	Decrease in BMD 0.22 Z-score/year
Roehhl et al. [70]	Modified ketogenic diet 15 gm carb vs. 50 gm carb diet	Constipation (9%)
Lambrechts et al. [71]	KGD	GI side effects (30%)

Abbreviations: KGD—ketogenic diet; MAD—modified Atkins diet; MCT—medium chain triglyceride diet; NR—non-report; LGID—low glycemic index diet.

## Data Availability

The data presented in this study are available in this article.

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
