# Peer review of "Incidence and Characteristics of Kidney Stones in Patients on Ketogenic Diet: A Systematic Review and Meta-Analysis"

_diseases, 2021, doi:10.3390/diseases9020039_

Round 1

Reviewer 1 Report

The review by Prakrati Achraya et al. deals with a very interesting and current topic. Several authors have already reported in the literature the increased risk of developing kidney stones in subjects undergoing ketogenic diet therapy regimens, especially if they are long-lasting. The paper is well written and easily understood. The methodology applied is also adequate for the intended purpose.

However, in order to make it even more current, I suggest the authors to introduce some concepts and citations recently published and present in Pubmed (and not only here).

Ketogenic diet was originally proven to be effective in epilepsy, and long-term follow-up studies on epileptic children undergoing a ketogenic diet reported an increased incidence of bone fractures and decreased bone mineral density. However, the causes of such negative impacts on bone health have to be better defined (Energy Metabolism and Ketogenic Diets: What about the Skeletal Health? A Narrative Review and a Prospective Vision for Planning Clinical Trials on this Issue. Daniela Merlotti, Roberta Cosso, Cristina Eller-Vainicher, Fabio Vescini, Iacopo Chiodini, Luigi Gennari, and Alberto Falchetti. Int J Mol Sci. 2021 Jan; 22(1): 435). In particular, it has been reported that idiopathic osteoporosis and nephrolithiasis, show a progressive increase in their incidence and prevalence, observed in both pediatric and adult populations worldwide. Epidemiological and experimental studies indicate that both disorders show several common pathogenic environmental and genetic factors. Probably, genetic predisposition and environmental factors may be recognized as triggers in adult and pediatric ages (Idiopathic Osteoporosis and Nephrolithiasis: Two Sides of the Same Coin? Rendina D, De Filippo G, Iannuzzo G, Abate V, Strazzullo P, Falchetti A. Int J Mol Sci. 2020 Oct 31;21(21):8183. doi: 10.3390/ijms21218183).

The populations that today can use ketogenic diet regimens are not only limited to pediatric ones, with drug-resistant epilepsy or GLUT1 gene alteration, nor only obese and/or type 2 diabetic or PCOS subjects and so on, but even athletes, both professionals and amateur. These last categories can also suffer damage to skeletal health and/or phospho-calcium metabolism also linked to abundant loss of calcium with sweating, especially if subjected to endurance training exercises and if no adequately supplemented with calcium salts (JAMA. 1996;276:226-230).

In obese subjects, the use of a VLCKD also provides, in the first phase of ketosis, a restriction of milk and dairy products which, in addition to containing lactose, represent the main sources of calcium for our body and its restriction, particularly in subjects already predisposed to fragility fracture risk, may determine a negative calcium balance with impaired bone strength, especially if on a prolonged diet (Energy Metabolism and Ketogenic Diets: What about the Skeletal Health? A Narrative Review and a Prospective Vision for Planning Clinical Trials on this Issue. Daniela Merlotti, Roberta Cosso, Cristina Eller-Vainicher, Fabio Vescini, Iacopo Chiodini, Luigi Gennari, and Alberto Falchetti. Int J Mol Sci. 2021 Jan; 22(1): 435).

In the studies considered in this, most likely, there was no mention of how many potentially could be cases of kidney stones in which there was a family predisposition, as well as environmental exposure, nutrition, physical activity, exposure to different sunlight. Are there any studies, among these, that reported familiarity for hypercalciuria? familiarity with hypocitraturia?

These concepts are important because they must induce careful clinical observation of subjects who are prescribed a ketogenic diet, possibly trying to identify subjects already potentially at stone risk before starting it.

In fact, approximately 40-45% of patients with idiopathic hypercalciuria have at least one other family member who has or has had calcium nephrolithiasis. Hypocitraturia, generally defined as urinary citrate excretion less than 320 mg per day for adults, is a common metabolic abnormality in stone formers, occurring in 20% to 60% and single nucleotide polymorphism in the gene encoding NaDC-1, which may be associated with reduced urinary citrate excretion in recurrent stone formers (Curhan GC, Taylor EN. 24-h uric acid excretion and the risk of kidney stones. Kidney Int. 2008; Chow K, Dixon J, Gilpin S, et al. Citrate inhibits growth of residual fragments in an in vitro model of calcium oxalate renal stones. Kidney Int. 2004; Nicar MJ, Skurla C, Sakhaee K, Pak CY. Low urinary citrate excretion in nephrolithiasis. Urology. 1983; Rudman D, Kutner MH, Redd SC 2nd, et al. Hypocitraturia in calcium nephrolithiasis. J Clin Endocrinol Metab. 1982; Pak CY. Etiology and treatment of urolithiasis. Am J Kidney Dis. 1991; Pak CY, Poindexter JR, Adams-Huet B, Pearle MS. Predictive value of kidney stone composition in the detection of metabolic abnormalities. Am J Med. 2003;115:26-32; Okamoto N, Aruga S, Matsuzaki S, et al. Associations between renal sodium-citrate cotransporter (hNaDC-1) gene polymorphism and urinary citrate excretion in recurrent renal calcium stone formers and normal controls. Int J Urol. 2007).

I suggest to mention that most of these did not predict, as primary outcome, the correlation between levels of KBs, in blood and / or urine, and stone risk.

Finally, I suggest citing the paper “Genetic variants for personalized management of very low carbohydrate ketogenic diets. Lucia Aronica, Jeff Volek, Angela Poff, Dominic P D'agostino” in anticipation of being able to implement in the early future precision medicine approaches also in the diet therapy field.

Author Response

Response to Reviewer#1

Comment #1

The review by Prakrati Achraya et al. deals with a very interesting and current topic. Several authors have already reported in the literature the increased risk of developing kidney stones in subjects undergoing ketogenic diet therapy regimens, especially if they are long-lasting. The paper is well written and easily understood. The methodology applied is also adequate for the intended purpose.

However, in order to make it even more current, I suggest the authors to introduce some concepts and citations recently published and present in Pubmed (and not only here).

Ketogenic diet was originally proven to be effective in epilepsy, and long-term follow-up studies on epileptic children undergoing a ketogenic diet reported an increased incidence of bone fractures and decreased bone mineral density. However, the causes of such negative impacts on bone health have to be better defined (Energy Metabolism and Ketogenic Diets: What about the Skeletal Health? A Narrative Review and a Prospective Vision for Planning Clinical Trials on this Issue. Daniela Merlotti, Roberta Cosso, Cristina Eller-Vainicher, Fabio Vescini, Iacopo Chiodini, Luigi Gennari, and Alberto Falchetti. Int J Mol Sci. 2021 Jan; 22(1): 435). In particular, it has been reported that idiopathic osteoporosis and nephrolithiasis, show a progressive increase in their incidence and prevalence, observed in both pediatric and adult populations worldwide. Epidemiological and experimental studies indicate that both disorders show several common pathogenic environmental and genetic factors. Probably, genetic predisposition and environmental factors may be recognized as triggers in adult and pediatric ages (Idiopathic Osteoporosis and Nephrolithiasis: Two Sides of the Same Coin? Rendina D, De Filippo G, Iannuzzo G, Abate V, Strazzullo P, Falchetti A. Int J Mol Sci. 2020 Oct 31;21(21):8183. doi: 10.3390/ijms21218183).

Response: We thank you for reviewing our manuscript. We agree with the reviewer’s point, and appreciate the reviewer’s expertise. We have incorporated these very important points in the discussion and added the references listed above as new references 80 and 81. The following text has been added in the discussion.

Epidemiological studies have shown a temporal relationship between idiopathic osteo-porosis and kidney stones. In addition, changing dietary patterns including ketogenic diet could possibly be an important environmental trigger in the association as well [88]. Bone health should be monitored closely in patient on ketogenic diet and more clinical trials are needed to further define the negative impacts on bone health [89].”

Comment #2

The populations that today can use ketogenic diet regimens are not only limited to pediatric ones, with drug-resistant epilepsy or GLUT1 gene alteration, nor only obese and/or type 2 diabetic or PCOS subjects and so on, but even athletes, both professionals and amateur. These last categories can also suffer damage to skeletal health and/or phospho-calcium metabolism also linked to abundant loss of calcium with sweating, especially if subjected to endurance training exercises and if no adequately supplemented with calcium salts (JAMA. 1996;276:226-230).

Response: We highly appreciate the reviewer’s suggestions and have added these important points in the introduction and discussion along with the citations (new reference 7). The following text has been added in the discussion.

Although prophylactic calcium and vitamin D is recommended in all people on ketogenic  diet for bone health [22,23].,athletes with dermal calcium loss during exercise / sweating and obese subjects restricting dairy, are at further risk of worsening bone health, if not on adequate calcium supplements [7, 89].

Comment #3

In obese subjects, the use of a VLCKD also provides, in the first phase of ketosis, a restriction of milk and dairy products which, in addition to containing lactose, represent the main sources of calcium for our body and its restriction, particularly in subjects already predisposed to fragility fracture risk, may determine a negative calcium balance with impaired bone strength, especially if on a prolonged diet (Energy Metabolism and Ketogenic Diets: What about the Skeletal Health? A Narrative Review and a Prospective Vision for Planning Clinical Trials on this Issue. Daniela Merlotti, Roberta Cosso, Cristina Eller-Vainicher, Fabio Vescini, Iacopo Chiodini, Luigi Gennari, and Alberto Falchetti. Int J Mol Sci. 2021 Jan; 22(1): 435).

Response: We agree with this comment. We have reviewed and added this very informative point in our discussion section, and have added these important points in the introduction and discussion along with the citations (new references 89).  

Comment #4

In the studies considered in this, most likely, there was no mention of how many potentially could be cases of kidney stones in which there was a family predisposition, as well as environmental exposure, nutrition, physical activity, exposure to different sunlight. Are there any studies, among these, that reported familiarity for hypercalciuria? familiarity with hypocitraturia?

These concepts are important because they must induce careful clinical observation of subjects who are prescribed a ketogenic diet, possibly trying to identify subjects already potentially at stone risk before starting it.

Response: Above is a very important suggestion but unfortunately the reviewed studies did not have data regarding above points, so could not be added. This is a limitation of our metanalysis and thus has been added with the limitations in the discussion section. The following text has been added in the discussion.

Our study has the following limitations. First, the observational studies included in the analysis are susceptible to shortcomings inherent to the design. In addition, sources of heterogeneity could be due to differences in patient population and definitions of ketogenic diet as described in Table 2. Second, important risk factors such as family history of nephrolithiasis, physical activity, exposure to sunlight, presence of ketone bodies in blood or urine and environmental exposure were not mentioned in the included studies.”

Comment #5

In fact, approximately 40-45% of patients with idiopathic hypercalciuria have at least one other family member who has or has had calcium nephrolithiasis. Hypocitraturia, generally defined as urinary citrate excretion less than 320 mg per day for adults, is a common metabolic abnormality in stone formers, occurring in 20% to 60% and single nucleotide polymorphism in the gene encoding NaDC-1, which may be associated with reduced urinary citrate excretion in recurrent stone formers (Curhan GC, Taylor EN. 24-h uric acid excretion and the risk of kidney stones. Kidney Int. 2008; Chow K, Dixon J, Gilpin S, et al. Citrate inhibits growth of residual fragments in an in vitro model of calcium oxalate renal stones. Kidney Int. 2004; Nicar MJ, Skurla C, Sakhaee K, Pak CY. Low urinary citrate excretion in nephrolithiasis. Urology. 1983; Rudman D, Kutner MH, Redd SC 2nd, et al. Hypocitraturia in calcium nephrolithiasis. J Clin Endocrinol Metab. 1982; Pak CY. Etiology and treatment of urolithiasis. Am J Kidney Dis. 1991; Pak CY, Poindexter JR, Adams-Huet B, Pearle MS. Predictive value of kidney stone composition in the detection of metabolic abnormalities. Am J Med. 2003;115:26-32; Okamoto N, Aruga S, Matsuzaki S, et al. Associations between renal sodium-citrate cotransporter (hNaDC-1) gene polymorphism and urinary citrate excretion in recurrent renal calcium stone formers and normal controls. Int J Urol. 2007).

Response: We highly appreciate the reviewer’s expertise in the topic and have reviewed more studies and added the above suggestions on risk factors for nephrolithiasis and in the discussion section to further make the paper more informative. . We have incorporated these very important points in the discussion and added the references listed above as new references 75, 76, 77, 78, 79, 81, and 82.

Comment #6

I suggest to mention that most of these did not predict, as primary outcome, the correlation between levels of KBs, in blood and / or urine, and stone risk.

Response: This is a very important point. The reviewed studies did not have this variable included and thus data could not be collected on levels of KB. This is a limitation in our meta-analysis and has been added there as suggested.

Comment #7

Finally, I suggest citing the paper “Genetic variants for personalized management of very low carbohydrate ketogenic diets. Lucia Aronica, Jeff Volek, Angela Poff, Dominic P D'agostino” in anticipation of being able to implement in the early future precision medicine approaches also in the diet therapy field.

Response: We agree. We highly appreciate drawing our attention towards this very informative paper and have added this point and referenced it (reference 90) in the discussion section.

We greatly appreciated the reviewer’s and editor’s time and comments to improve our manuscript. The manuscript has been improved considerably by the suggested revisions.

Reviewer 2 Report

This paper provides a review of the incidence of kidney stones in patients on ketogenic diet.

The revision of relevant articles has a good strategy, and the analysis of included articles is well performed. The results and the discussion of the analyzed articles are correct and consistent with what was expected

In this sense, this is an interesting work, although the paper cannot be classified as an original article, since, as indicated in the title it is a systematic review.

The paper merits to be published as a Review.

Author Response

Response to Reviewer#2

Comment #1

 This paper provides a review of the incidence of kidney stones in patients on ketogenic diet.

The revision of relevant articles has a good strategy, and the analysis of included articles is well performed. The results and the discussion of the analyzed articles are correct and consistent with what was expected

In this sense, this is an interesting work, although the paper cannot be classified as an original article, since, as indicated in the title it is a systematic review.

The paper merits to be published as a Review.

Response:  We greatly appreciated the reviewer’s and editor’s time and comments to improve our manuscript. The manuscript has been improved considerably by the suggested revisions.